# Dynamic Features of Chromosomal Instability during Culture of Induced Pluripotent Stem Cells

**DOI:** 10.3390/genes13071157

**Published:** 2022-06-27

**Authors:** Casey O. DuBose, John R. Daum, Christopher L. Sansam, Gary J. Gorbsky

**Affiliations:** 1Cell Cycle and Cancer Biology Research Program, Oklahoma Medical Research Foundation, Oklahoma City, OK 73104, USA; cdubose@uchicago.edu (C.O.D.); john-daum@omrf.org (J.R.D.);; 2Department of Cell Biology, University of Oklahoma Health Sciences Center, Oklahoma City, OK 73104, USA

**Keywords:** aneuploidy, mitosis, trisomy, chromosome 12, optical mapping, structural variants, insertions, deletions, translocations

## Abstract

Induced pluripotent stem cells (iPSCs) hold great potential for regenerative medicine. By reprogramming a patient′s own cells, immunological rejection can be avoided during transplantation. For expansion and gene editing, iPSCs are grown in artificial culture for extended times. Culture affords potential danger for the accumulation of genetic aberrations. To study these, two induced pluripotent stem (iPS) cell lines were cultured and periodically analyzed using advanced optical mapping to detect and classify chromosome numerical and segmental changes that included deletions, insertions, balanced translocations and inversions. In one of the lines, a population trisomic for chromosome 12 gained dominance over a small number of passages. This appearance and dominance of the culture by chromosome 12 trisomic cells was tracked through intermediate passages by the analysis of chromosome spreads. Mathematical modeling suggested that the proliferation rates of diploid versus trisomic cells could not account for the rapid dominance of the trisomic population. In addition, optical mapping revealed hundreds of structural variations distinct from those generally found within the human population. Many of these structural variants were detected in samples obtained early in the culturing process and were maintained in late passage samples, while others were acquired over the course of culturing.

## 1. Introduction

Since their creation in 2007, induced pluripotent stem cells (iPSCs) have offered great promise in the field of regenerative medicine. By utilizing four key transcription factors, Octomer 3/4 (Oct-3/4), SRY-box containing gene 2 (Sox2),Krüppel-like factor 4 (Klf4) and cytoplasmic Myc (c-Myc), the Yamanaka group was able to induce differentiated somatic cells to a pluripotent state [1]. These iPSCs display similar characteristics to embryonic stem cells (ESCs), namely, the ability to self-renew and differentiate into a wide range of somatic cells. iPSCs, however, have the potential to provide an alternative source of pluripotent cells while avoiding possible immunological rejection because they can be derived from the patient being treated. However, to be used in this manner, iPSCs must be created and expanded in culture. The requirement for growth in culture opens the possibility of the accrual of chromosomal variants over time. 

Candidate cells must be cultured under artificial conditions during the reprogramming process and expanded after they are successfully transformed into iPSCs and potentially used for gene editing. During this extended period of cell culture, mutations, chromosome abnormalities or DNA damage may arise [2,3,4,5,6,7,8,9,10,11,12,13]. These changes may promote the iPSCs to become tumorigenic and develop other genetic or epigenetic abnormalities that make them risky for therapeutic use [4,14,15,16,17]. Interestingly, the acquisition of a transformed phenotype in a subpopulation of stem cells may drive neoplastic gene expression and phenotype changes in surrounding normal stem cells [18].

A variety of genomic abnormalities can affect stem cells in culture. One of the most significant genome changes is aneuploidy [19,20,21]. An investigation of over 200 iPSC lines found that 12.5% of the cultures examined had an abnormal karyotype, while a study of 125 ESC lines found that 34% of cell lines contained abnormal karyotypes; both studies demonstrate the widespread occurrence of chromosomal aberrations [12,22]. In particular, chromosome 12 in iPSCs has been shown to have a high propensity for trisomy, representing as much as 32% of all chromosome aberrations detected in iPSCs [9,12]. The accrual of multiple chromosome 12p arms has also been shown to repeatedly occur in human embryonal carcinoma cells [23]. The appearance of large chromosome changes correlated with the altered expression of genes on the chromosome. Lines with an additional 12p segment overexpress pluripotency genes, *Homeobox NANOG* (*NANOG)* and *Growth Differentiation Factor 3* (*GDF3)* [9]. Smaller segmental changes, termed structural variants, are comprised of deletions, insertions, inversions, duplications and translocations of at least 50 base pairs [24]. These may also have a significant impact on gene expression. In the overall human genome, on average, single nucleotide polymorphisms (SNPs) contribute to 0.1% genetic variation between individuals, while structural variants contribute 1.5% [25]. 

A wide variety of disorders have been associated with structural variants, including cancer, diabetes and cognitive disease [26,27]. Structural variants implicated in disease can affect a single gene coding region, multiple genes, or can affect gene regulators at a distance [28,29]. Although structural variants create significant genetic diversity and are implicated in various diseases, they are difficult to discover using short-read sequencing and therefore remain poorly mapped compared to SNPs. One solution to this problem is long-read sequencing, which aims to produce reads of thousands of base pairs, thereby easing the process of mapping and increasing structural variant sensitivity.

In this study, we examine the effects of cell culture on the genomic integrity of iPSCs by culturing two related induced pluripotent stem (iPS) cell lines in parallel for 50 passages. The cell lines were examined at various time points throughout the experiment by optical mapping supplemented by chromosome counts. Optical mapping creates marked DNA fragments that can be assembled into whole genomes quickly and efficiently [30]. This technology can detect chromosome and structural variants with high accuracy. We detected hundreds of structural variants in both iPS cell lines, including many not previously mapped as existing human alleles. We identified both preexisting variants in these lines and those acquired during culture. We documented the gain of an additional chromosome 12 in one line. Of the structural variants acquired over the course of culturing, many disrupted protein coding sequences. 

## 2. Results and Discussion

### 2.1. IPSC Growth Rate and Aneuploidy 

A concern with multiple passages of any cell line in culture is that some cells may acquire genetic changes that allow favorable adaptation to the specific culture conditions [14,31]. An advantageous mutation may increase the proliferation rate or decrease the rate of apoptosis, eventually leading the variant progeny to become dominant in the cell population [32,33,34]. Our study involved two human iPSC lines. The first line was denoted WTC-11 iPSC [35], and was the cell line from which the second iPSC line, AICS-0012, was derived (Available online: https://www.allencell.org/cell-catalog.html (accessed on 20 May 2022)). AICS-0012 contains an mEGFP tag on the N-terminus of the coding sequence of the a-tubulin gene TUBA1B, created with CRISPR-Cas9 technology at the Allen Cell Institute, and is referred to in this study as Tuba1-GFP iPSC (Available online: https://www.allencell.org/cell-catalog.html (accessed on 20 May 2022)). Note that for both WTC-11 iPSC and Tuba1-GFP iPSC, passage numbering reflects the number of passages performed during the course of this study, starting with passage number zero and ending with fifty. By using two iPSC lines from the same donor, we avoided potential differences in gene expression that may have arisen from differences in the individual donors [36]. We counted the number of living cells present during each passage of both iPSC lines, allowing us to monitor the doubling times of both cell lines over time (Figure 1). Neither cell line demonstrated significant changes in doubling time over the course of the experiment, indicating that genetic variants with reduced doubling time did not overtake the lines during the culture periods. One benefit of our chosen optical mapping platform, the Bionano Genomics Saphyr, was its ability to detect copy number changes throughout a sample′s genome down to the subchromosomal level. By analyzing the same cell line at multiple time points, this capability allowed us to examine the change in the gene dosage of the cell population, potentially identifying adaptations to cell culture conditions. One significant change was the gain of an additional chromosome 12 in Tuba1-GFP iPSC line, first identified in passage 32 via optical mapping (Figure 2).

As noted above, trisomy 12 is a common aneuploidy in iPSC lines and ESC lines [6,12,21,22,23,37]. Conventionally this has been hypothesized to occur through increased expression of growth promoting genes that provide a growth advantage to the variants [6,14,23]. However, none of the four samples analyzed (excepting the copy number increase in the Tuba1-GFP iPSC Passage 32 trisomy) displayed an increase in copy number or structural variation events, such as duplications for *NANOG* or *GDF3*. In an early comprehensive study examining 69 hESC lines and 37 hiPSC lines, large duplications, including some trisomies of chromosomes 12, 17 and 20, were observed multiple times [38]. Several hESC lines showed a conserved small duplication on chromosome 12 of a region just adjacent to *NANOG* containing *NANOGP1*, a *NANOG*-related pseudogene. Another broad study of ESC and iPSC lines identified several common trisomies, including chromosome 12, but generally reported that genes, such as *NANOG*, do not disproportionately acquire structural variations during cell culture [22]. That study also suggested that the small duplication adjacent to *NANOG* may simply reflect a pre-existing polymorphism present in the human population. 

Whatever changes of expression occur, one common explanation for the dominance of the chromosome 12 trisomic population is that cells with this trisomy have a growth advantage, allowing them to outgrow the normal diploid cells. However, we did not detect a significant change in rate of proliferation after the trisomy became dominant. A recent publication reported that certain hESC chromosome variant lines could actively induce apoptosis in cells of the parental line in mixed cultures [34]. However, again, this mechanism of overgrowth required that the chromosomal variant lines have a significantly higher proliferation rate [34]. We did not detect trisomies or large structural changes in the WTC-11 line, even at later passages. While this might be explained simply by stochastic variation, an alternative explanation could be that the Tuba1-GFP iPSC line is more susceptible to aneuploidy due to the additional time spent in culturing conditions, necessitated by the GFP tagging process. We obtained the WTC-11 line from the Coriell Institute at absolute passage 39, which included passages of the original fibroblasts obtained from the donor, reprogramming and expansion of the line. The Allen Cell Institute obtained the WTC-11 line at passage 33 and passaged it an additional 32 times to generate, clone and expand the Tuba1-GFP line. Thus, compared to the WTC-11 line we obtained, the Tuba1-GFP iPSC line spent approximately 26 more passages under culturing conditions at the time we received it. During each passage, we froze samples, allowing us to reexamine chromosome content in more detail at later times. 

To map the acquisition of trisomy in the Tuba1-GFP iPSCs, we thawed selected passages and performed chromosome spreads, focusing on the passages during which the trisomy arose and became dominant (Figure 3). 

Tuba1-GFP iPSC population shifts from 46, the expected number for diploid human cells, to 47 over the course of 5 passages. The cell population containing the trisomy is initially modest, reflecting only 12.5% at passage 21, which may reflect, at least in part, the error rate inherent in counting chromosome spreads. However, the trisomic population rapidly increases to 80% by passage 26. The aberrant genotype then persists in the cell population, as indicated by 84% of the population possessing 47 chromosomes at passage 40. By combining this information with the recorded doubling time information, we can approximate that the trisomy 12 genotype shifted from 12.5% to 80% of the population in approximately 20 doublings. This is a surprisingly short time period, given that the post-trisomy 12 population does not divide at a significantly faster rate compared to the diploid population. The doubling times for the predominantly diploid Tuba1-GFP cultures, passages prior to passage 21, and for the predominantly chromosome 12 trisomy cultures, passages post passage 26, though not found to be statistically significantly different, were determined to be 18.3 ± 1.8 and 17.9 ± 1.1, respectively.

To understand the potential mechanisms that might explain the rapid increase in chromosome 12 trisomy, we created a series of mathematical models based on the measured growth of the early passage diploid cells and late passage trisomic Tuba1-GFP cells depicted in Figure 1. We compared the conditions in which the initial population consisted of 10% trisomic cells. Based on the measured growth rates and simple competition, we calculated that, after approximately 360 h occurring during 5 passages, the proportion of trisomic cells would only rise to 12.2% (Figure 4A). In contrast, experimental observations derived from chromosome spreads indicated that the proportion of trisomic cells rose from ~10% to ~85% over the same duration of 5 passages (passages 21 through 26, Figure 2). We then modulated other parameters to model potential drivers for the rapid increase in trisomy from 10 to 90% in 360 h. We determined that trisomic cells would have to exhibit a doubling time of approximately 12 h, rather than the 18 h observed, and thus proliferate at approximately 1.5 times the rate of the diploid cells to achieve 90% culture dominance in 360 h (Figure 4B). For preferential cell death of diploid cells to account for the change, our model indicates that, to achieve rapid dominance, the diploid cells would have to show 20% cell death per division while the trisomic cell death rate would be 0% (Figure 4C). Although we have not yet specifically tested differences in cell death, this level of difference would have been unlikely to escape our notice during cell culture. Finally, we modeled the situation in which chromosome missegregation leading to trisomy was not restricted to a single founder cell, but could occur in multiple diploid cells during each division. We found that, to account for the change in dominance, the missegregation would require that 10% of the diploid cells would become trisomic for chromosome 12 during each division (Figure 4D). Because we have stocks of cells frozen during the critical passages, we are currently investigating whether any of these scenarios, or indeed others not yet modeled, may account for the rapid rise in trisomic cells. 

### 2.2. Structural Variant Detection, Filtration and Characterization

We analyzed four samples via Bionano Genomics (BNG) optical mapping, two for each cell line. These were taken once early in an early passage and once in a late passage. Optical mapping in this way works by labeling a single known sequence on large fragments of genomic DNA, often larger than 150 kbp. Labeled fragments are then aligned to create a de novo assembly which, for human samples, is compared to a known reference map. Doing so allows for the detection of structural variants, some of which may contain repeated sequence and be missed in next generation sequencing. All samples exceeded recommended molecule quality requirements, and the metrics for each sample can be seen in Appendix A. Each sample was assembled de novo and then mapped to human reference hg38 using BNG solve (version 3.4) and access software (version 1.4.2). Structural variant calls were filtered with recommended thresholds excepting minimum required molecule coverage, which was doubled. To identify structural variants unique to our iPSC samples, we filtered our results to exclude all structural variants found in the BNG control database, which comprised samples from the 1000 Genome Project and donors from San Diego Blood Banks. In the four samples, we identified 169 deletions, 81 insertions, 47 duplications and 97 inversions, all of which were absent from the 1000 genomes project low-frequency alleles database, as determined by using the Ensembl. Variant Effect Predictor [39,40]. Given that the four cell populations analyzed came from two cell lines that both share a common ancestor, we examined how many variants were present in 2 or more samples. A variant was considered unique if it had less than a 50% reciprocal overlap with any other variant. Of the total 394 variants detected, 233 (59%) were present in more than one sample. The distribution of structural variants across all samples can be seen in Figure 5.

Both WTC-11 and Tuba1-GFP iPSC lines demonstrated maintained, lost and acquired structural variants from early to late passage samples. The total structural variant frequency increased in both lines over the course of culturing. Inversion calls from both lines showed a low rate of maintained variants compared to other structural variant types. However, this may be due to detection limits from the reduced sensitivity of the optical mapping technology for inversions below 30 kbp. The size distribution of structural variants varied by variant type and cell line, and can be seen in Figure 6. The chromosome locations and sizes for maintained, lost and acquired variants are indicated in Appendix A. Among both cell lines, duplications were on average the largest structural variants, inversions an intermediate size, while deletions and insertions were similar in size and smaller than other types of structural variants.

### 2.3. Structural Variation Impact on Gene Function

Acquired structural variations can impart negative or, perhaps more rarely, positive consequences for cell survival. If a structural variant completely supplants a gene, the loss or gain in copy number can directly affect expression. Changes in gene expression may have negative downstream consequences, impacting expression of other genes or compromising proteostasis through the over- or under-expression of proteins normally produced as balanced components of protein complexes [41,42,43,44]. Similarly, if a structural variant encroaches partially into protein coding sequences of a gene, truncations or gene fusions may arise, leading to a loss or gain of function [28,45,46]. In order to understand the potential consequences, both in the genome and phenotypically, we used the Ensembl Variant Effect Predictor to predict the impact of our structural variant set [40]. To identify potentially high-impact effects of our detected structural variants, we focused directly on protein coding consequences and identified 121 genes whose coding sequence overlaps with structural variants from the four samples (Figure 7). In the case of both cell lines, the sample collected later in the culture experiment contained more genes whose protein coding sequences were affected by structural variants. Of the 121 genes affected, 28 were present in both cell lines, perhaps owing to their common parental line. Interestingly, one gene, DAPL1, which was unaffected in the early passage samples of both cell lines, was modified in late passage samples by an insertion variant into an intron segment of the protein coding region. DAPL1 has been implicated in epithelial differentiation, apoptosis and suppressor of cell proliferation in the retinal pigment epithelium [47]. Notably, DAPL1 has been identified as a potential susceptibility locus for age-related macular degeneration in females [48]. 

Then, we utilized the functional annotation toolset DAVID to determine if the structural variant impacted genes disproportionately affected particular gene ontology groups or were linked to any disease associations [49]. Gene lists from each sample, shown in Figure 7, were divided into two groups—genes affected by duplications and thus potentially enriched, and non-duplication structural variants that could potentially interfere with gene expression. In the WTC-11 iPSC line, many gene ontology groups were disrupted by deletions, insertions and inversions. The most statistically significant disruptions were driven by SVs acquired over the course of culturing. Conversely, no gene ontology groups were significantly affected by duplications (Figure 8A). The Tuba1-GFP iPSC line was impacted by both disruptive and duplicative SVs. Similar to WTC-11, the most statistically significant impacts arose from SVs, which were acquired during culture (Figure 8B). Notably, large structural variants can impact several genes, which may lead to a particular gene ontology term being deemed enriched if the neighboring genes are homologues or perform similar functions. For instance, the gene group “gonadal differentiation” in the WTC-11 passage 45 sample was likely enriched due to six genes on the Y chromosome being impacted by a single large deletion (Figure 8A).

## 3. Prospects and Conclusions

Different methods for passaging iPSCs and ESCs are in common use, including mechanical dissection and transfer of colonies, or generation and transfer of small clumps and single cells generated by divalent cation chelation or proteolytic enzymes [50]. The temporary application the ROCK inhibitor to cultures after passaging was introduced to promote the attachment of cells during passaging [51]. Passage methods and the use of the ROCK inhibitor may affect the expression of stem cell genes [52]. Both WTC-11 and Tuba1-GFP iPSC lines were previously conditioned to enzymatic passaging, as clumps or single cell suspension, respectively. We followed protocols reported by the Allen Cell Institute, which included single cell suspension and the use of ROCK inhibitor. Both enzymatic passaging and a common alternative, mechanical passaging of colonies, have been associated with promoting genome aberrations, though single cell enzymatic passaging has been the more strongly implicated [53,54,55,56]. Enzymatic single cell passaging remains a standard technique for passaging stem cells. It is essential for clinical applications as well as gene editing and clonal creation of homogenous cell lines.

Over the course of 150 days of continuous culture, or 50 passages, we observed a substantial change in the genomes of two iPSC lines. Most notably, the Tuba1-GFP iPSC cell line experienced the appearance and rapid dominance of a population trisomic for chromosome 12. Trisomy 12 is a well-documented aneuploidy amongst stem cells. However, the mechanism by which a cell population becomes dominated by such a karyotype remains unclear [9,12,23,57]. It has been suggested that trisomy 12 leads to a proliferative or selective advantage in artificial culture due to a change in gene dosage, particularly the pluripotency gene NANOG [9,19,23], reviewed in [5,21,58]. However, the insignificant change in proliferation rate following the amplification of chromosome 12 observed during our culturing implicates a mechanism more complicated than simple growth advantage. The processes that guide aneuploidy gain and maintenance are still not fully understood. This may be particularly true for iPSCs and other pluripotent cells. Interestingly, the correction of trisomy to diploidy has been reported during or after reprogramming of trisomic patient cells [59,60]. One explanation to explain this phenomenon is that higher proliferation rates in diploid cells, compared to trisomic cells, compromised by proteostasis or other defects, could allow overgrowth by the diploid population after a spontaneous loss of trisomy during mitosis. 

In our study, given that the proliferation rate of the cells did not significantly increase after acquiring the trisomy, it is unlikely that the rapid dominance of the trisomic population can be attributed to simple competition or differences in cell death rates. Further studies actively pursuing the mechanism or mechanisms underlying the rapid conversion of the population may uncover a novel source of stem cell aneuploidy.

We detected hundreds of structural variants not found in the general population using long-read optical mapping technology. However, 59% of structural variants were found in more than one sample, suggesting that those variations may be due to the unique genetic constitution present in the donor genome. More significantly, both iPSC lines acquired numerous structural variants over the course of culturing. After ascertaining the genes whose protein coding sequences were affected by structural variants, we were able to identify several enriched gene ontology and disease clusters. While it is unclear if these changes might compromise use of iPSCs in therapy, the accumulation of variants suggests that culture times be minimized in therapeutic practice. 

## 4. Methods

### 4.1. IPS Cell Lines

The parental cell line, WTC-11 iPSC, was derived from human male fibroblast cells using episomal vectors (OCT3/4, short hairpin p53 siRNA (shp53), SOX2, KLF4, MYCL Proto-Oncogene (LMYC) and LIN28 Homolog (LIN28) [35]. A frozen cell aliquot was obtained from the Coriell Institute (identifier GM25256) at passage 39 and cultured for this study (passages 1–50). Whole genome sequencing and population level RNA-seq data for the WTC-11 iPSC line are publicly available from the Allen Cell Institute/UCSC Genome Browser.

The modified IPS cell line derived from the WTC-11 line, denoted Tuba1-GFP in this study, is part of the Allen Cell Collection (identifier AICS-0012). Using CRISPR/Cas9, WTC-11 iPSCs at passage 33 were endogenously tagged with mEGFP in a single allele of the TUBA1B gene and then cloned (clone 105) at the Allen Cell Institute. Tuba1-GFP cells were obtained from the Coriell Institute (identifier AICS-0012) at passage 32 from the Allen Cell Collection and then cultured in this study (passages 1–50).

### 4.2. Cell Culture

Induced Pluripotent Stem Cells (iPSCs) were cultured in 25 square centimeter flasks coated in Growth Factor Reduced Matrigel (Corning 354230) diluted in DMEM/F12 at 1:30 ratio (Caisson Labs DFL14–500ML). Flasks were coated with Matrigel and used within one week, with care taken to avoid evaporation or drying of the Matrigel solution. Cells were maintained with mTesR1 and mTesR plus media (Stemcell Technologies, Vancouver, BC, Canada, 85850 and 05825) supplemented with penicillin-streptomycin (Thermo Fisher Scientific, Waltham, MA, USA, 15-140-122). With each passage, 300,000 iPS cells were transferred to a new flask following detachment via Accutase (Thermo Fisher Scientific, Waltham, MA, USA, A1110501). Culturing media was supplemented with 10 uM Y-27632 ROCK inhibitor (MedChem Express, Monmouth Junction, NJ, USA, HY-10583) for approximately 24 h following passaging, then changed to media without ROCK inhibitor. Cells were maintained at 37C in 5% CO_2_ in a water-jacketed incubator. Cells were passaged approximately every 72 h and were maintained continuously for 150 days. iPSC colonies maintained normal morphology with minimal differentiation and had an average death rate of approximately three percent.

### 4.3. Chromosome Spreads

Cells were treated with Nocadazole at a concentration of 100 ng/mL for 4 h, then treated with Accutase to bring cells into suspension. Cells were collected, then washed with media by centrifuging at 200× *g* for 3 min. Cells were resuspended in 500 µL of warmed swelling buffer (70% deionized water + 30% mTesR media). Cells were incubated in a 37 °C water bath for 20 min. Cells were then fixed by adding 1 mL of freshly prepared 3:1 methanol to acetic acid and then incubated at room temperature for 15 min. Cells were centrifuged for 5 min at 200× *g*, washed with 1 mL of fixative and pelleted again. The cells were resuspended in 150–200 µL of fixative, then 50–60 µL of cell suspension was dropped from a height of 70 cm onto a 22 mm^2^ coverslip. The coverslips were then placed inside a 150 mm dish on top of wet filter paper. The coverslips were then allowed to dry overnight. Next, the coverslips were stained with 4′, 6-diamidino-2-phenylindole (DAPI) (100 ng/mL) and SYBERGold nucleic acid dye (Thermo Fisher Scientific Waltham, MA, USA, S11494) at a 1:20,000 dilution of stock. Imaging was performed with a Zeiss Axioplan II microscope platform using a 100× objective, Hamamatsu Orca II camera and Metamorph software.

### 4.4. Predictive Mathematical Modeling

A mathematical model created using Microsoft Excel predicted fractions of cells within culture containing a mixed population of two cell types, Tuba1-GFP diploid and Tuba1-GFP chromosome 12 trisomy, exhibiting unique characteristics. Inputs permitted cell-type-specific manipulation of doubling times, cell death rates and conversion rates from one cell type to another, so that theoretical cell type fractions within the culture could be predicted over time with respect to these varying input characteristics. For probing cell-type-specific death rates, the formulae employed a percentage loss of newly created daughter cells at each doubling time. Conversion rates were calculated similarly, with certain percentages of diploid cells converting to chromosome 12 trisomy cells at each diploid doubling time. Microsoft Excel stacked area graphs were utilized to provide illustrations of predictive results.

## 5. Bionano Genomics Techniques

### 5.1. Genomic DNA Isolation

iPS cells were collected from culture and counted using the Countess II FL Automated Cell Counter (Thermo Fisher Scientific, Waltham, MA, USA, AMQAF1000). 1 × 10^6^ and 1.5 × 10^6^ cell aliquots (corresponding to 6 and 9 µg of DNA) were targeted for each sample preparation. Samples were prepared immediately after being collected, following Bionano guidelines (Bionano Genomics, Bionano Prep Cell Culture DNA Isolation Protocol, Doc. 30026). Briefly, cells were pelleted, resuspended in cold Cell Buffer (Bionano Genomics, San Diego, CA USA) and suspended in agarose plugs in order to minimize DNA shearing (Bio-Rad, Hercules, CA, USA, CHEF Mapper XA system, 1703713). The DNA-agarose plugs were subjected to a series of Proteinase K Digestions (Qiagen, Hilden Germany 158920) followed by an RNase A digestion (Qiagen, Hilden, Germany 158922). Following digestion treatments, the plugs were washed with 1x Wash Buffer (Bionano Genomics, San Diego, CA, USA) and Tris-EDTA Buffer (Invitrogen, Waltham, MA, USA, AM9849). The DNA-agarose plugs were then digested with agarase (Thermo Fisher Scientific, Waltham, MA, USA, EO0461) at 43C for 45 min. The DNA solution was then purified via drop dialysis, using Millipore filters floated on TE buffer for 1 h. Following purification, the DNA was quantified through the use of a Qubit 4 Fluorometer (Thermo Fisher Scientific, Waltham, MA, USA, Q33238). DNA with concentrations of 35–200 ng/µL and a coefficient of variation (standard deviation/mean) of less than 0.25 was deemed acceptable. 

### 5.2. DNA Labeling

DNA labeling was achieved using the Direct Label Staining (DLS) method from Bionano Genomics (Bionano, San Diego, CA, USA, Prep Direct Label and Stain Protocol, 30206 D), whose protocol instructions were strictly followed. Briefly, 750 ng of purified genomic DNA was added to a master mix including the DLE-1 Labeling Enzyme and incubated for 2 h at 37 °C. Subsequently, the sample was treated with Proteinase K (Qiagen, Hilden, Germany, 158920) and incubated at 50 °C for 30 min. Then, the samples were cleaned up via membrane adsorption using reagents supplied by Bionano Genomics. Following cleanup, the labeled DNA was stained and homogenized in preparation for loading the sample onto the Bionano chip. After staining, the samples were incubated at room temperature overnight. The following day the stained DNA samples were quantified using a Qubit 4 Fluorometer and its complimentary dsDNA High-Sensitivity Assay Kit. Samples with concentrations between 4 and 12 ng/µL were chosen to be loaded onto the Bionano Chip. 

### 5.3. Chip Loading and Analysis

Once the labeled DNA was quantified, it was loaded onto a Bionano Saphyr chip. The chip was then loaded into the Saphyr instrument where the labeled DNA was linearized into nanochannels using electrophoretic principles to guide the DNA. Once loaded, the Saphyr instrument began imaging the labeled DNA. Each flow cell can process 320–480 Gbps of DNA in the span of 24 to 36 h. The data output was analyzed using Bionano Solve 3.4. The compiled data were then visualized through Bionano Access software, version 1.4.2.

### 5.4. Structural Variant Analysis Software

Structural variant analysis involving co-localization with other detected structural variants and comparing variant coordinates across samples to determine if the structural variants were “maintained”, “lost” or “unique” was performed using bed-tools intersect (version 2.26.0). Structural variant impact, such as effect on protein coding genes, was determined using the Ensembl Variant Effect Predictor (VEP) [40]. Gene ontology analysis of structural variants was performed using the Database for Annotation, Visualization and Integrated Discovery (DAVID) version 6.8 [49].

### 5.5. Statistical Analysis

Statical analysis related to iPSC line doubling times (Figure 1 and Figure 3) was performed using GraphPad prism software unpaired t-tests. Significance of gene ontology group enrichment (Figure 8) was generated using DAVID webtools (version 6.8). Star values represent EASE scores, which represent modified Fisher’s exact *p*-values (* = *p* ≤ 0.05, ** = *p* ≤ 0.01, *** = *p* ≤ 0.001).

## Figures and Tables

**Figure 1 genes-13-01157-f001:**
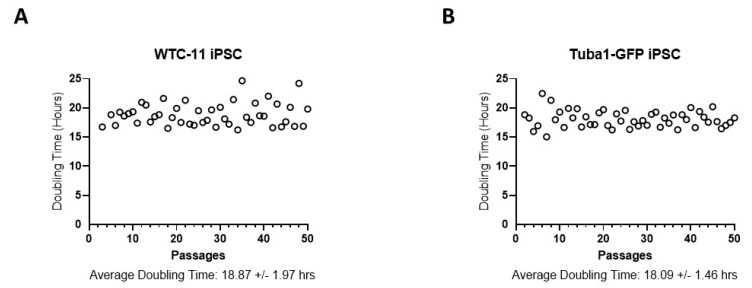
**Doubling times of iPSCs vary little during culturing.** (**A**) WTC-11 cell line. (**B**) Tuba1-GFP cell line. Cell counts were obtained with each passage using an automated cell counter, and the time between each passage was recorded in order to calculate population doubling times. Passage numbers reflect documented passages following the creation of WTC-11 iPSC line and begin at the passage number for which they were thawed for these experiments. There was no significant change in doubling times between the first ten and last ten passages in either cell line (unpaired *t*-test).

**Figure 2 genes-13-01157-f002:**
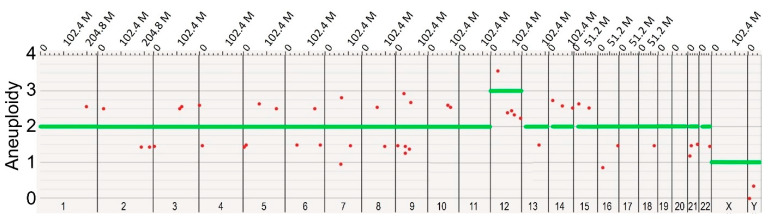
**The late passage of Tuba1-GFP iPSC displays a gain of chromosome 12.** Optical mapping data of Tuba1-GFP iPSC passage 32 detected a copy number of three for chromosome 12 in the majority of cells, indicating an acquired aneuploidy. The green horizontal bars indicate the ploidy of each chromosome in the sample population, while the red dots indicate specific subchromosomal regions that vary in copy number from the called ploidy of the chromosome. In order to call a whole chromosome aneuploidy, it is necessary for 80% or more of the chromosome length to be categorized as higher or lower than the copy number baseline. Note that the X and Y chromosomes report a copy number of one, as the sample is derived from a male donor. Data visualized using Bionano Access software.

**Figure 3 genes-13-01157-f003:**
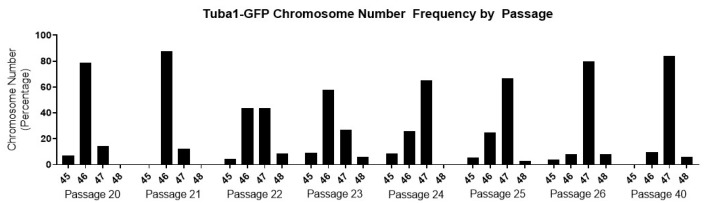
**The presence of an aneuploidy event in the Tuba1-GFP iPSC population is confirmed via chromosome spreads**. Each sample reflects the ploidy of the passage population. Passages 22 to 26 indicate a progressive shift from normal ploidy to aneuploidy as the chromosome numbers of the populations shift from 46 to 47. The aneuploidy of the population then persists in the population, as indicated in the passage 40 sample. Each passage reflects >20 cells quantified.

**Figure 4 genes-13-01157-f004:**
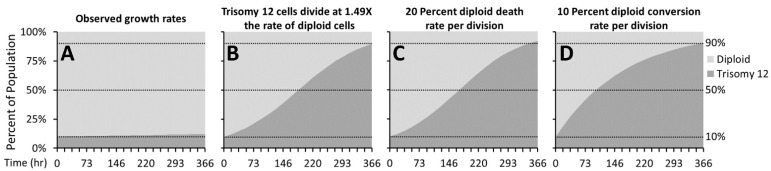
**Mathematical modeling of various growth parameters**. Mathematical simulations modeling the effects of various parameters of the conversion of iPSCs from predominantly diploid to predominantly trisomy 12 populations in the 20 doublings that occur during 5 passages. (**A**) Simulation using the observed growth rates of majority diploid iPSCs (passages 1–21) and majority trisomy 12 iPSCs (passages 26–50) fails to model the rise in trisomy 12 dominance. (**B**) Modeling trisomy 12 cells replicating with doubling time of 12.5 h, at 1.49 times the rate of diploid cells. (**C**) Modeling a very high (20%) death rate for diploid cells and a very low (1%) rate for trisomy 12 cells. (**D**) Modeling our hypothesized mechanism of a 10% conversion rate of diploid to trisomy 12 cells per doubling.

**Figure 5 genes-13-01157-f005:**
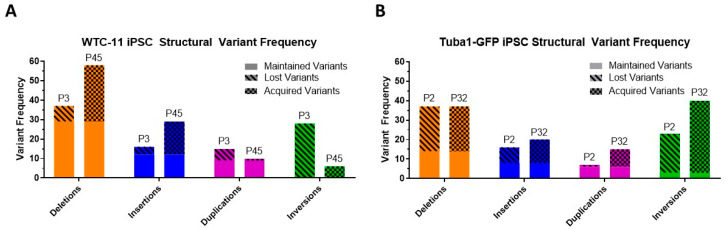
**Structural variant frequency varies from early passage sample to late passage in both iPSC lines**. Using optical mapping, two samples from each cell line were analyzed, one early passage sample and one late passage sample. Structural variants were organized into three categories: maintained variants, which were detected in both early and late passage samples; lost variants, which were detected in the early passage samples, but not in the late passage samples; and acquired variants, which were not detected in the early passage samples, but subsequently appeared in the late passage samples. (**A**) Structural variant frequencies from WTC-11 iPSC passages 3 and 45 are shown. Deletions and insertions demonstrate an increase in frequency from early to late passage samples, while total duplications and inversions decrease. (**B**) Structural variant frequencies from Tuba1-GFP iPSC passages 2 and 32 are shown. All variant types exhibit an increase in frequency from early to late passage samples with the exception of deletions, which maintain the same total instances.

**Figure 6 genes-13-01157-f006:**
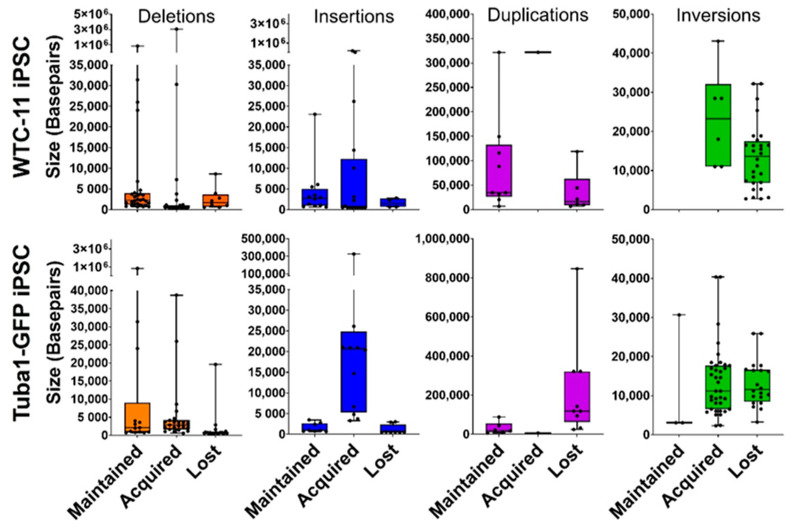
**Structural variants vary in size distribution**. All detected structural variants are represented on box and whisker plots, where the box indicates the first, median and third quartiles. Notably, duplications were the least frequent and on average largest structural variants in both iPSC lines.

**Figure 7 genes-13-01157-f007:**
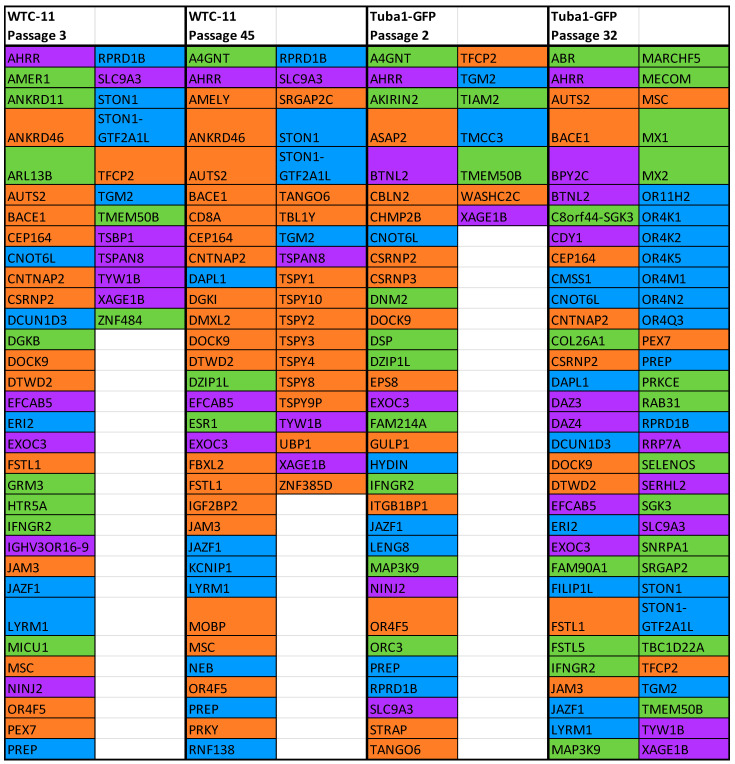
**Structural Variants Affect Coding Sequences of Many Genes**. List of genes whose protein coding sequences are impacted by structural variants. Impacts may be serve, such as complete deletion or truncation, or may be more moderate, such as intron variants. Deletions, insertions, duplications and inversions are indicated by orange, blue, purple and green, respectively.

**Figure 8 genes-13-01157-f008:**
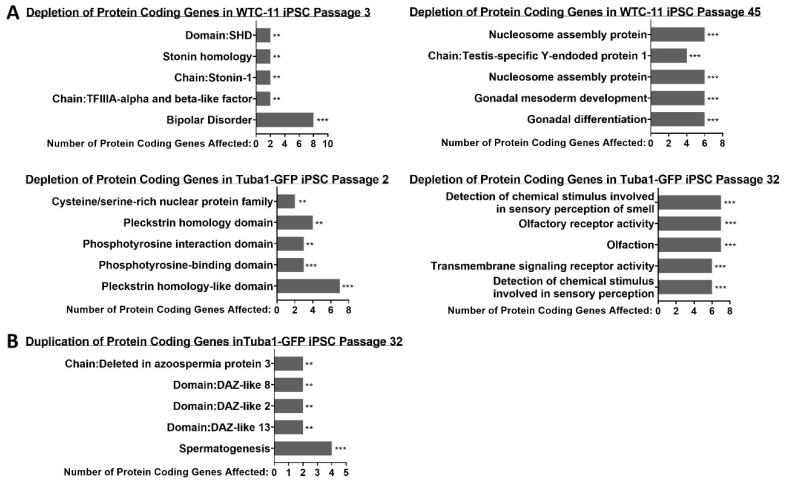
**Many gene ontology groups are disproportionately affected by structural variants**. (**A**) Many gene ontology groups were depleted as a consequence of a deletion, insertion or inversion interfering with the coding sequence of a gene. (**B**) Only the late passage sample of the Tuba1-GFP iPSC line displayed a significant enrichment of gene ontology groups affected by duplication structural variants, which can lead to the overexpression of the gene product. Asterisks reflect EASE scores, which represent modified Fisher’s exact *p*-values (** *p* ≤ 0.01, *** *p* ≤ 0.001).

## Data Availability

Bionano optical mapping files .cmap, and bnx data can be found via the NCBI Supplementary Files database; BioProject submission SUB10829253. https://submit.ncbi.nlm.nih.gov/subs/supfiles/SUB11453665, accessed on 20 May 2022; https://submit.ncbi.nlm.nih.gov/subs/supfiles/SUB11453657, accessed on 20 May 2022; https://submit.ncbi.nlm.nih.gov/subs/supfiles/SUB11453648, accessed on 20 May 2022; https://submit.ncbi.nlm.nih.gov/subs/supfiles/SUB10823566, accessed on 20 May 2022.

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
