# Peer review of "Dynamic Features of Chromosomal Instability during Culture of Induced Pluripotent Stem Cells"

_genes, 2022, doi:10.3390/genes13071157_

Round 1
Reviewer 1 Report
Major issue:
a- The number of cell lines’ passages before the study is critical information and should be included (50 during the study);
Minor issues:
a- Define Oct-4, Sox2, Klf4, and c-Myc genes in the Introduction when they are first cited;
b- remove the word “numerical” from line 2, page 2, and maybe change “aneuploidy” for “euploidy”;
c- define the genes s NANOG and GDF3 (page 2);
d- what is “(cite)” at the beginning of the Results section?
e- ESC has already been determined; remove it from line 6, page 4
f- check a few typos throughout the text.
Author Response
We thank the reviewers for helping us improve the manuscript. Below we reproduce the reviewers’ comments and provide our responses underlined.
Reviewer 1
Open Review
(x) I would not like to sign my review report
( ) I would like to sign my review report
English language and style
( ) Extensive editing of English language and style required
( ) Moderate English changes required
(x) English language and style are fine/minor spell check required
( ) I don't feel qualified to judge about the English language and style
|
Yes |
Can be improved |
Must be improved |
Not applicable |
|
|
Does the introduction provide sufficient background and include all relevant references? |
(x) |
( ) |
( ) |
( ) |
|
Are all the cited references relevant to the research? |
(x) |
( ) |
( ) |
( ) |
|
Is the research design appropriate? |
(x) |
( ) |
( ) |
( ) |
|
Are the methods adequately described? |
(x) |
( ) |
( ) |
( ) |
|
Are the results clearly presented? |
(x) |
( ) |
( ) |
( ) |
|
Are the conclusions supported by the results? |
(x) |
( ) |
( ) |
( ) |
Comments and Suggestions for Authors
Major issue:
a- The number of cell lines’ passages before the study is critical information and should be included (50 during the study);
We have added complete details of the passage histories of the two lines before we started passaging them in our laboratory (page10, lines 30 – 43)
Minor issues:
a- Define Oct-4, Sox2, Klf4, and c-Myc genes in the Introduction when they are first cited;
Done. (page1, lines 30-31)
b- remove the word “numerical” from line 2, page 2, and maybe change “aneuploidy” for “euploidy”;
We removed the modifier “numerical” before “aneuploidy” in the instance noted by the reviewer and once again later (page 4, line 26). However, we do not believe substituting “euploidy” for “aneuploidy” is appropriate since we are referencing chromosome abnormalities.
c- define the genes s NANOG and GDF3 (page 2);
Done. (page2, lines 14-15)
d- what is “(cite)” at the beginning of the Results section?
This was a missing citation which has been added.
e- ESC has already been determined; remove it from line 6, page 4
done
f- check a few typos throughout the text.
We have searched for and corrected typos.
Reviewer 2 Report
The manuscript is overall comprehensive and well written however, some minor issues should be addressed.
1. In results and discussion, up-to-date references are not provided and few of the references does not appear to be exhaustive.
2. Indeed, there is a wealth of published data showing that ESCs and iPSCs apparently acquire genetic changes/variations in long term culture; when observed these variants in perspective: many PSC lines acquire variations in late passage especially when adapted to conditions like single cell passaging. However, the process of single cell passaging is highly essential for scaling-up the PSC cultures for drug screening or regenerative medicine. Adapting to single-cell passaging with ROCKi can be compromise the integrity of the iPSCs and can result in the appearance of genomic abnormalities, such as trisomy 12 (Bai et.al 2015). As the single-cell passaging is always a “debated technique” why the authors preferred this method rather than using collagenase or EDTA-based clump passaging methods; which is a more widely popular technique. Hence, the authors should elaborate on this statement by adding further context.
Author Response
We thank the reviewers for helping us improve the manuscript. Below we reproduce the reviewers’ comments and provide our responses underlined.
Reviewer 2
Open Review
(x) I would not like to sign my review report
( ) I would like to sign my review report
English language and style
( ) Extensive editing of English language and style required
( ) Moderate English changes required
(x) English language and style are fine/minor spell check required
( ) I don't feel qualified to judge about the English language and style
|
Yes |
Can be improved |
Must be improved |
Not applicable |
|
|
Does the introduction provide sufficient background and include all relevant references? |
( ) |
(x) |
( ) |
( ) |
|
Are all the cited references relevant to the research? |
( ) |
( ) |
(x) |
( ) |
|
Is the research design appropriate? |
(x) |
( ) |
( ) |
( ) |
|
Are the methods adequately described? |
(x) |
( ) |
( ) |
( ) |
|
Are the results clearly presented? |
(x) |
( ) |
( ) |
( ) |
|
Are the conclusions supported by the results? |
(x) |
( ) |
( ) |
( ) |
Comments and Suggestions for Authors
The manuscript is overall comprehensive and well written however, some minor issues should be addressed.
- In results and discussion, up-to-date references are not provided and few of the references does not appear to be exhaustive.
- Indeed, there is a wealth of published data showing that ESCs and iPSCs apparently acquire genetic changes/variations in long term culture; when observed these variants in perspective: many PSC lines acquire variations in late passage especially when adapted to conditions like single cell passaging. However, the process of single cell passaging is highly essential for scaling-upagethe PSC cultures for drug screening or regenerative medicine. Adapting to single-cell passaging with ROCKi can be compromise the integrity of the iPSCs and can result in the appearance of genomic abnormalities, such as trisomy 12 (Bai et.al 2015). As the single-cell passaging is always a “debated technique” why the authors preferred this method rather than using collagenase or EDTA-based clumpagepassaging methods; which is a more widely popular technique. Hence, the authors should elaborate on this statement by adding further context.
We have added 28 additional references to the 32 present in the original manuscript. Many of these are comprehensive and recent analyses concerning the accumulation of genetic variants during culturing of ESCs and iPSCs.
We have also added discussion and references comparing the effects of different culture methods on accumulation of variants (page 9, lines 17-30). As pointed out by the reviewer the single cell passaging is essential in some instances for drug screening and for derivation and selection of gene-edited variants.
Reviewer 3 Report
The manuscript by Casey O. Dubose and colleagues focusses on genetic genetic aberrations occurring in long-term cultures iPSCs. The Authors studied two lines taking advantage of optical mapping to detect and classify chromosome numerical and segmental changes that included deletions, insertions, balanced translocations and inversions. They found a population trisomic for chromosome 12 and hundreds of structural variations distinct from those generally found within the human population.
Overall, the manuscript is interesting and clearly written but it suffers from some minor issues. Here are some examples:
1. It is essential to add information on the reprogramming method used to obtain iPSCs (those involving the integration of exogenous genetic material and those involving no genetic modification of the donor cells?) and on the type of somatic cell from which they were obtained. Furthermore, there is a lack of information on the number of clones used to carry on the described experiments. It is possible that there is also variability among clones, so please provide this information.
2. The Authors should provide a small paragraph in the "Methods" section related to statistical analysis.
3. The Authors should enrich the bibliography with the most recent papers concerning the topic of chromosomal aberrations on iPSCs model.
I also suggest extending the "Conclusions" paragraph with few sentences about the relevance of the study to improve its clarity, conciseness, impact, and risks of using this model in regenerative medicine.
Author Response
We thank the reviewers for helping us improve the manuscript. Below we reproduce the reviewers’ comments and provide our responses underlined.
Reviewer 3
Open Review
(x) I would not like to sign my review report
( ) I would like to sign my review report
English language and style
( ) Extensive editing of English language and style required
( ) Moderate English changes required
(x) English language and style are fine/minor spell check required
( ) I don't feel qualified to judge about the English language and style
|
Yes |
Can be improved |
Must be improved |
Not applicable |
|
|
Does the introduction provide sufficient background and include all relevant references? |
( ) |
(x) |
( ) |
( ) |
|
Are all the cited references relevant to the research? |
( ) |
( ) |
(x) |
( ) |
|
Is the research design appropriate? |
( ) |
(x) |
( ) |
( ) |
|
Are the methods adequately described? |
( ) |
( ) |
(x) |
( ) |
|
Are the results clearly presented? |
( ) |
(x) |
( ) |
( ) |
|
Are the conclusions supported by the results? |
( ) |
( ) |
(x) |
( ) |
Comments and Suggestions for Authors
The manuscript by Casey O. Dubose and colleagues focusses on genetic aberrations occurring in long-term cultures iPSCs. The Authors studied two lines taking advantage of optical mapping to detect and classify chromosome numerical and segmental changes that included deletions, insertions, balanced translocations and inversions. They found a population trisomic for chromosome 12 and hundreds of structural variations distinct from those generally found within the human population.
Overall, the manuscript is interesting and clearly written but it suffers from some minor issues. Here are some examples:
- It is essential to add information on the reprogramming method used to obtain iPSCs (those involving the integration of exogenous genetic material and those involving no genetic modification of the donor cells?) and on the type of somatic cell from which they were obtained. Furthermore, there is a lack of information on the number of clones used to carry on the described experiments. It is possible that there is also variability among clones, so please provide this information.
Characterization of the parental and derived iPSCs has been added to the manuscript (page10, lines 30-43)
- The Authors should provide a small paragraph in the "Methods" section related to statistical analysis.
This paragraph has been added (page12, lines 33-38)
- The Authors should enrich the bibliography with the most recent papers concerning the topic of chromosomal aberrations on iPSCs model.
We have added 28 references to the 32 that were present in the original manuscript. Many of these concern the topic of chromosome aberrations in iPSCs.
I also suggest extending the "Conclusions" paragraph with few sentences about the relevance of the study to improve its clarity, conciseness, impact, and risks of using this model in regenerative medicine.
We have expanded the “Conclusions” paragraph to “Prospects and Conclusions” addressing various issues regarding the use of iPSCs and their importance in research and regenerative medicine (bottom of page 9 and top of page 10).